# OpenReview forum: "Towards a Unified View of Neuron Interpretation and Behavior Control in Large Language Models"
_ICLR.cc/2026/Conference — ICLR 2026 Conference Withdrawn Submission_

### Official Review · Reviewer_dJSr · 2025-10-28

**Soundness:** 2
**Presentation:** 3
**Contribution:** 3
**Rating:** 4
**Confidence:** 4

**Summary:**

The paper proposes a linear framework intended to unify interpretability and controllability in large language models. It collects hidden activations from model responses to selected example prompts and treats each collected activation vector as a “concept,” forming a concept matrix. It then learns a linear decoder that maps any new hidden state at a chosen layer into coefficients over these concepts.  Those coefficients indicate which behaviours are active in that state, and that modifying the coefficients and reconstructing the activation lets them steer the model’s behaviour at inference time.

**Strengths:**

The paper offers a single linear pipeline that links interpretation and control, which is conceptually clean.

The reconstructed activations can be swapped in with little or no perplexity increase and extremely low reconstruction error, suggesting the method can intervene without obviously breaking the model.

The approach exposes a more human facing control surface than typical steering methods: instead of adjusting opaque neuron groups or unlabeled latent directions, the user is effectively turning specific behaviours up or down using concepts derived from natural-language examples

**Weaknesses:**

The notion of a “concept” is weakly grounded. Instead of learning disentangled latent factors from activations (as in dictionary learning / SAE work), the method simply takes raw activation vectors from prompted examples and declares each one to be a concept. The sentences are likely to be polysemantic bundles of behaviour, not clean features. Calling this “concept discovery” or “interpretability” stretches the standard meaning of those terms in the cited literature.

The reported errors seem high for modern SAEs. Since the overall objective is different (see previous point); therefore, it does not make sense to lean into it.


The experiments regarding steering are narrow and should be expanded to evaluate a broader set of behaviours in benchmarks such as AXBENCH, which the authors have cited.

**Questions:**

Please clarify how the SAE reconstruction error is computed in Table 1? Are these normalised per-dimension/per-token average?

---

> ### Author Response · Authors · 2025-11-23
> **Response to the reviewer dJSr**
>
> Thanks for appreciating our work and we would like to express our deepest gratitude to thank the reviewer for the constructive feedback!
>
> **Weakness 1: Notion of concept is weakly grounded and sentences may not be clean features.**
>
> (1) The notion of what constitutes a "clean" concept lacks rigorous definition in the literature. Concepts can always be further decomposed into more granular sub-concepts without clear boundaries. Besides, there is actually no limit on the number of concepts that human can conjure [ref1]. Among all these concepts, it is hard to define whether one concept is "cleaner" than another concept.
>
> [ref1] Rajendran G, et al. From causal to concept-based representation learning. NIPS2024
>
> (2) Even if individual sentences contain multiple semantic elements, their comparative use can isolate specific concepts. For example, contrasting sentences containing concepts "A+B+C" with those containing "A+B" effectively isolates concept "C". This contrastive approach is particularly valuable for complex concepts like "refusal of harmful prompts" that are crucial for AI safety but difficult to define directly without examples. Our framework allows such contrastive uses by increasing and decreasing coefficients of a pair of sentences.
>
> (3) We are not inventing a concept discovery mechanism aiming to discover "ground-truth" concepts. Our framework acknowledges the subjective nature of conceptual organization and provides a flexible way for users to understand models through their preferred conceptual lenses. We believe acknowledging the fact that human can understand the same thing in arbitrarily large number of possible ways is important to further advance the design of interpretation frameworks. As a simple example: a concept "car" can be decomposed into concepts like "chasis", "wheels", "engine", but can also be decomposed into concepts like "metal", "plastic", "rubber", or "control system", "safety system", "comfort system", etc. Every person may come up with a different way to understand what is a "car".
>
> (4) Interpreting concepts in SAE is also based on comparing the concept representations with the activations extracted from sentences [line 95-98]. This indicates that prior SAE based frameworks also acknowledge that raw activations of sentences do deliver interpretable semantic meanings.
>
> Overall, we hope our above point of view could be considered as a contribution to the interpretability community in setting a more human-centered perspective in the interpretation framework design, instead of a weakness.
>
> **Weakness 2: High reconstruction error in SAE due to different overall objective; does not make sense to lean into it.**
>
> (1) As an interpretation framework, both SAE and our unified framework do (and should) commonly aim to achieve low neuron reconstruction error because this metric indicates how much information is not explained by the interpretation framework [line 340].
>
> (2) After our improvement, directly using SAE concepts in our interpretation framework also yields precise neuron reconstruction, as shown in Table 4. So we compare with original SAE frameworks to justify our framework's benefits in neuron reconstruction.
>
> (3) Other design "objective" in our framework such as user-friendly concept specification and controllability are inherent merits of our framework and are not directly related to neuron reconstruction errors.
>
> We agree that our design objective is not completely the same as SAE. However, since we follow SAE's methodology in understanding neurons via decomposing the neuron into interpretable concepts, and the design of our framework is based on the observation of the existing flaws of SAE based frameworks [line 167-174], we believe SAE frameworks are (at least the best possible) relevant baselines we could compare with.
>
> **Weakness 3: Evaluate more behaviors in benchmarks such as AXBENCH.**
>
> In AXBENCH, different behaviors are evaluated via different generated synthetic data based on different concepts. The steering vectors are also calculated based on synthetic data from different behaviors. However, how the steering vectors are computed (e.g., "DiffMean" and "PCA" approach in AXBENCH) are just the same as what we introduce in section 4.2. So the performance in behavior control would just be the same, as our framework's performance guarantee does not rely on any specific dataset.
>
> Since currently we encounter some issues in running the code released by AXBENCH, we do not have new results for now. We will try our best to gain some results before the end of the rebuttal period.
>
> **Question 1: How SAE reconstruction errors are calculated.**
> They are averaged across dimensions but cumulative across different activations/neurons [line 343]. This error metric is applied to all methods including different SAE variants and our framework.
>
> We hope the above responses could address the reviewer's concerns and sincerely hope the reviewer could consider raising the score. Thanks!

---

> ### Author Response · Authors · 2025-11-29
> **Follow-up of the response to weakness 3: Evaluate more behaviors in benchmarks such as AXBENCH.**
>
> We expand the steering evaluation to 5 more behaviors of a representative approach "Diffmean" in AXBENCH. This approach can be unified into our framework as introduced in section 4.2.
>
> **AXBENCH task introduction** The behaviors that AXBENCH evaluates are of the following form: after giving certain instructions to the steered LLM, a judge-LLM judges whether a specified concept is incorporated into the response of the steered LLM when that concept's corresponding steering vectors are added during the inference. Such instructions are sampled from Alpaca-eval [ref1].
>
> [ref1] Li X, Zhang T, Dubois Y, et al. Alpacaeval: An automatic evaluator of instruction-following models[EB/OL].(2023-5)
>
> **AXBENCH score introduction** The steering score is calculated via a harmonic mean of the fluency score, instruction-following score and concept-presence score, which are evaluated via the judge LLM. Fluency score represents how fluent the response is. Instruction-following score represents how well the response is related to the instruction. Concept-presence score indicates how well the concept is incorporated into the response. All scores are judged using 3 discrete values via the judge-LLM: 0,1,2 (0 is the worst and 2 is the best score).
>
> The "harmonic mean" used in AXBENCH is calculated as follows: (1) if any score of the 3 scores is 0, the harmonic mean is 0. (2) Otherwise it's calculated as
>
> $\frac{3}{1/fluen+1/inst+1/concept}$,
>
> where "fluen", "inst", "cocnept" indicates 3 types of scores, respectively.
>
>
> **Evaluated behavior/concepts** The base model we use is Gemma-2-2b-inst and the intervention is conducted in layer 20 following AXBENCH. The corresponding concepts of the 5 behaviors are included in CONCEPT500 of AXBENCH. These 5 concepts are:
>
> (1) Statements or phrases involving the act of saying or expressing something.\
> (2) Statements about the nature and condition of entities.\
> (3) Biographical information about a person.\
> (4) References to different worlds, realities, or fantastical settings within narratives.\
> (5) Technical vocabulary related to chemical processes and materials synthesis.
>
> **New results** The scores in the below table is the average of harmonic score of all 37 sampled instructions in each behavior, respectively. The intervention magnitude is set to $\alpha=1$ in all experiments for a fair comparison. We conduct 3 runs and report detailed results below:
>
>
> |Concept/Behavior index | 1 |2 |3 |4|5|
> |--|--|--|--|--|--|
> |**Diffmean**|| || | |
> |Run1|0.291667	|0.4916	|0.2055	|0.0888	|0.1166|
> |Run2|0.4500|	0.6055	|0.2667|	0.0667|	0|
> |Run3|0.4667	|0.4889	|0.2806|	0.0750|	0|
> |Mean| 0.4028	|0.5287|	0.2509|	0.0769	|0.0389|
> |Variance|0.0010|	0.0035|	0.0002|	2.94 $\times 10^{-5}$ |	0.0005|
> |Overall mean of 5 behaviors | 0.25963| | | | |
> |**Our unified framework** | | | | | |
> |Run1|0.3667|0.6611|0.1917|0.0417|0.0417|
> |Run2|0.5806|0.6556|0.1917|0.075|0.0833|
> |Run3|0.5000|0.6389|0.2806|0|0.0417|
> |Mean|0.4824|0.6519|0.2213|0.0389|0.0556|
> |Variance|0.0117|	0.0001	|0.0026|	0.0014	|0.0006|
> |Overall mean of 5 behaviors |0.29| | | | |
>
> From the above table, it could be seen that **our unified framework can maintain the performance of the original method** (and it even performs slightly better) thanks to the theoretical guarantee our framework offers. The performance variance comes from the variance of the Judge-LLM (GPT-4o-mini)'s output as well as the floating point conversion during the process of computing $\mathbf{W}_{dec} \in \mathbb{R}^{c \times n}$ (c is the concept number and n is the channel dimension).
>
> This process requires the inverse operation and the torch.inverse() does not support data type bf16 originallly used in the Gemma2-2B-inst. So we convert the constructed concept matrix from bf16 into float64 to compute the $\mathbf{W}_{dec}$.
>
> Since the code in AXBENCH adopts float32 in the inference of steering, we further convert the $\mathbf{C}\mathbf{W}_{dec}$ to float32 to obtain the above results. The above series of floating point conversions yield precision loss and cause a minor performance difference. Note that theoretically the performance should be the same after being unified into our framework, as proved in section 4.2 (e.g.,
>
> $\mathbf{CW}_{dec}\mathbf{x} =\mathbb{I}\mathbf{x}=\mathbf{x}$ always holds for any neuron $\mathbf{x}\in \mathbb{R}^n$).
>
>
>
> We sincerely hope that the above response could resolve the reviewer's concern. Thanks!

---

### Official Review · Reviewer_ig9f · 2025-10-30

**Soundness:** 2
**Presentation:** 2
**Contribution:** 3
**Rating:** 6
**Confidence:** 3

**Summary:**

This paper introduces a unified framework to integrate the interpretation of neurons with the control of behavior in LLMs. The goal is to overcome the shortcomings of current Sparse Autoencoder-based interpretability frameworks (e.g., imprecise reconstruction, uninterpretable bias terms and ineffective control) while providing the explanatory power often absent in pure behavioral control techniques. Experimental results demonstrate that the SVR features significantly outperform baseline methods such as Sparse Autoencoders (SAE) and Attention Steering (AS) across multiple behavioral control tasks.

**Strengths:**

1.Clear and important motivation: This paper clearly articulates a crucial gap in current LLM neuron interpretability and controllability research. The proposed vision of a unified framework holds significant academic merit.
2.Methodological Novelty: The unified framework is novel and insightful. By changing the autoencoder's objective function—from reconstructing raw activations (SAE) to reconstructing task-specific steering vectors—it successfully integrates the supervisory signal of the control task into the feature learning process, which is the key to achieving highly controllable features.
3.Comprehensive Interpretability Analysis: The paper extends beyond control efficacy by providing robust interpretability analyses, including attribution analysis of SVR features and the identification of "Steerable Neurons" linked to specific behaviors (e.g., safety refusal), thereby fulfilling its promise of a "unified view."

**Weaknesses:**

1.This optimization problem solves for the L2-minimum norm solution for $W_{dec}$. L2 minimization tends to produce many small weights, not sparse weights. The paper claims this makes coefficients more selective and justifies this with only a single case study (0.1734 $\rightarrow$ -0.0028) in Section 5.3. This is weak. Why would the L2 solution systematically produce more interpretable coefficients than other solutions? The authors do not provide sufficient theoretical or experimental justification.
2.Infinity of solutions: As the paper acknowledges (Sec 4.1 and Appendix G), when the number of concepts $c > n$ , there are infinitely many combinations of $C$ and $W_{dec}$ that satisfy $C W_{dec} = \mathbb{I}$. The paper's choice of $W_{dec} = C^{T}(CC^{T})^{-1}$ is just one of these infinite solutions. This means the coefficients $W_{dec}x$ is entirely dependent on the user's choice of $C$ and this specific $W_{dec}$ solution. This seems more like a projection, projecting the neuron's activation into a user-specified concept basis, rather than truly discovering the model's internal computational mechanism. How do we know this explanation is what the model is really doing, versus a mathematically equivalent form we have imposed on it? This may conflict with the goal of neuron interpretability.
3.The framework's precise reconstruction relies on the number of concepts $c$ being greater than or equal to the neuron dimension $n$. For LLMs (e.g., Llama 3 8B, $n=4096$), this implies the user must provide at least 4096 (linearly independent) concept sentences. This is completely unrealistic in practice. The advantage of user-friendly and flexible specification becomes meaningless under the $c \ge n$ requirement.

**Questions:**

1.Can you provide broader evidence (except the case in Sec 5.3) that this L2 solution systematically yields more interpretable results than the other (infinitely many) solutions to $C W_{dec} = \mathbb{I}$?
2.The framework's guarantees rely on $c \ge n$. However, in a practical application, a user might only care about a small number of concepts (e.g., $c=10 \sim 20$), which is far less than $n=4096$. In this $c \ll n$ case (where $C W_{dec} = \mathbb{I}$ cannot be satisfied), how does the framework perform?
3.How sensitive is the entire interpretation to the specific choice of concepts in $C$? If I swap one "random" concept sentence in $C$ for another, do the interpretation coefficients change drastically? Does this imply the explanations themselves are unstable?

---

> ### Author Response · Authors · 2025-11-23
> **Response to the reviewer ig9f (Part 1/2)**
>
> Thanks for appreciating our work and we would like to express our deepest gratitude to thank the reviewer for the constructive feedback!
>
> **Weakness 1& Question 1: L2 norm only encourages small values, not sparse weights. Why more interpretable?**
>
> (1) There might be a misunderstanding here: although we call it "sparsity constraint", we enoucrage the sparsity regarding the concept coefficients
> $\mathbf{W}_{dec} \mathbf{x}$ [line 277],
> not the weight in
>
> $\mathbf{W}_{dec}$ itself. (we separate the sentence into 2 lines due to equation rendering problems in openreview. We separate sentences in below responses for the same reason.)
>
> (2) The reviewer is correct if we are directly minimizing the L2 norm of $\mathbf{W}_{dec}$.
>
> However, we are not minimizing the L2 norm of $\mathbf{W}_{dec}$.
>
> Instead, we are minimizing the L2 norm of $\mathbf{W}_{dec}-\mathbf{C}^T$.
>
> We do want to encourage many small values in $\mathbf{W}_{dec}-\mathbf{C}^T$
>
> such that $\mathbf{W}_{dec}$ would be close to $\mathbf{C}^T$.
>
> (3) Why our optimization makes $\mathbf{W}_{dec}\mathbf{x}$ sparse/more interpretable:
>
>  Intuitively, if any input $\mathbf{x}$ has a semantic meaning close/not close to any concept, human expects the corresponding coefficients to be large/small: the similarity between any neuron $\mathbf{x}$ and concepts $\mathbf{C}$ is expressed as $\mathbf{C}^T\mathbf{x}$,
> while the concept coefficients are expressed as $\mathbf{W}_{dec}\mathbf{x}$ [equation 6 and line 277].
>
> Therefore, it's straightforward to derive our optimization objective in equation 8 which encourages $\mathbf{W}_{dec}$ to be close to $\mathbf{C}^T$.
>
> (4) Other infinitely many possible solutions without our proposed constraint could yield large concept coefficients for concepts not similar to the input. Thus these solutions are systematically less interpretable.
>
> **Weakness 2: How do we know this explanation is what the model is really doing?**
>
> (1) In this work, we argue that seeking for a "ground-truth" concept set may not be that meaningful [Figure 6]. **First**, the meaning of "really used concept set" is hard to be rigorously defined: a LLM is mostly doing matrix/vector multiplications. As long as any matrix/vector can be precisely decomposed/reconstructed via certain concepts, the LLM's computation results would always be exactly the same. Any set of concepts achieving this could be considered as being "used" by the model. Thus it is hard to claim one concept set is "really used" and another concept set is not "really used". **Second**, as pointed out by [ref1], there is actually no limit on the number of concepts that human can conjure. In our context, this means human can come up with arbitrarily large number of possible concepts to understand a certain neuron if desired. As a simple example: a concept "car" can be decomposed into concepts like "chasis", "wheels", "engine", but can also be decomposed into concepts like "metal", "plastic", "rubber", or "control system", "safety system", "comfort system", etc. Every person may come up with a different way to understand what is a "car". Based on the above reasons, we do not seek for a "ground-truth" concept set that a model "really uses" and instead take a more human-centered perspective: decomposing a neuron's activation into a user specified concept basis. This philosophy is consistent with [ref2] which argues interpretable machine learning is exactly aiming to find one interpretable model among all possible models reaching the same performance [line 290-294].
>
> [ref1] Rajendran G, et al. From causal to concept-based representation learning[J]. NIPS2024
>
>
> [ref2] Rudin C. Stop explaining black box machine learning models for high stakes decisions and use interpretable models instead[J]. Nature machine intelligence, 2019, 1(5): 206-215.
>
> (2) For the same reason, none of the existing SAE based interpretation framework could justify the model is "really using" the discovered concepts neither. For example, given 2 SAEs reaching the same reconstruction error on a neuron, it is hard to claim the LLM is "really using" the concept set from one SAE while not "really using" the concept set from the other SAE.
>
> (3) We believe our framework indicates an important step forward in acknowledging the fact that there is no limit on the number of concepts that human can conjure [ref1]. Without acknowledging this fact, any interpretation framework may be endlessly making efforts to seek for a "really used" concept set and such a unique concept set may simply not exist. Note that interpretability research is exactly aiming to interpret to humans. Thus we believe taking a more human-centered perspective in interpretations is particularly meaningful, especially when the model's performance/precise neuron reconstruction can be guaranteed via our framework.

---

> ### Author Response · Authors · 2025-11-23
> **Response to the reviewer ig9f (Part 2/2)**
>
> **Weakness 3: Precise neuron reconstruction relies on the number of concepts exceeding neuron dimension, which is unrealistic for users.**
>
> Our framework has a unique advantage to allow users to write the concepts, but we do not impose users to manually write all of them: (1) users can leverage LLM to generate relevant concepts of interests or leverage any existing contrastive conversations proposed in any dedicated behavior control approaches. (2) Users can also simply use concepts from any trained SAE to construct the concept matrix in our framework. So it's not a huge burden for users to specify concepts but rather a flexibility in specifying them.
>
> Note that in original SAE based interpretation frameworks, even when the concept number is significantly larger than the neuron dimension, there is no guarantee of a precise neuron reconstruction .
>
> **Question 2: How the framework performs when users only care about a small set of concepts?**
>
> If users only wants to understand the neurons via $c<n$ concepts, a precise neuron reconstruction cannot be guaranteed. It might exist a large reconstruction error for a general neuron. However, this is not a design flaw of our framework. Instead, this is an insight our framework offers to indicate to which extend the user's specified concepts are not enough to explain the target neuron. Table 3 of the paper [line 459-464] shows the reconstruction and perplexity performance when using $c<n$ concepts in the pile-10k dataset.
>
> **Question 3: How sensitive is the entire interpretation to the specific choice of C? Does swapping one concept change interpretation coefficients drastically?**
>
> For a specific input neuron, swapping a concept highly related to this input has a larger influence on the coefficiets compared to swapping a concept less related to this input. This is an expected and reasonable explanation change.
>
> We fix 6000 random concepts and individually swap the 4 concepts shown in rows 1,2,4,6 in Figure 3 with a randomly generated unrelated concept. The below table shows how the coefficients change when the concept set is changed.
>
>
> |Coefficients of concepts | row1 | row2 | row4 | row6|
> |-------|------|------|------|-----|
> |Original coeff. | 0.2950 | 0.3363 | -0.0050 | 0.1945|
> |Swap row 4 concept (input unrelated)|0.3220 | 0.2810 | 0.0113 | 0.2096|
> |Swap row 1 concept (input related)| 0.0058 |0.4573|-0.0065|0.3027|
> |Swap row 2 concept (input related)| 0.4538 | 0.0053 | 0.0061 |0.2972|
> |Swap row 6 concept (input related)|0.4080 | 0.3423 | 0.0048 | 0.0150|
>
> The above results are consistent with our expectation where swapping more related concepts have a larger influence and swapping less related concepts have a minor influence.
>
> We further show how the coefficients of fixed related and unrelated concepts change when a large number of unrelated concepts are swapped: we fix the 4 concepts explicitly shown in Figure 3 (row1,2,4,6) while generating 6000 additional random concepts 3 times to simulate swapping 6000/6004 concepts to observe the coefficient changes of fixed 4 concepts shown in different rows of Figure 3.
>
> | Run | row 1 | row 2  | row 4  | row 6  |
> | ----|----|----|-----|----|
> |run1 | 0.2950 | 0.3363 | -0.0050 | 0.1945|
> | run2 | 0.3022  | 0.3308  | 0.0007  |0.1891|
> | run3 | 0.2913  |0.3332   | -0.0048   |0.1895|
> |run4 | 0.2979   |0.3344   | -0.0060   |0.1850|
>
> The above results show the coefficients of all concepts only have a minor change when irrelevant concepts are swapped even in large scale.
>
> All above results do not indicate anything related to whether an explanation is "stable". The concept set is the core part of flexibly understanding a neuron in a user-friendly manner. It is natural to expect an explanation change when the concept set is modified. Whether such a change is large depends on specific to-be-interpreted neuron and what concept the user wants to change. Note that in SAE based frameworks, changing the concept set can not be done in above user-friendly flexible manner and one have to train a new SAE network to obtain a new set of concepts. It's nearly impossible to expect "only one concept is swapped" in the new SAE network.
>
>
> We hope the above responses could address the reviewer's concerns and sincerely hope the reviewer could consider raising the score. Thanks!

---

### Official Review · Reviewer_rWXW · 2025-11-01

**Soundness:** 3
**Presentation:** 3
**Contribution:** 3
**Rating:** 6
**Confidence:** 3

**Summary:**

This paper proposes a unified framework aimed at bridging neuron interpretation and behavior control in Large Language Models (LLMs). The framework ensures that the decomposition of input neurons can be reconstructed precisely while achieving control effects comparable to existing behavior-control approaches. Empirical results demonstrate that the proposed method achieves lower reconstruction error and maintains equivalent behavior-control performance relative to prior work.

**Strengths:**

1. The paper is clearly presented and well-structured.
2. The idea of unifying neuron interpretation and behavior control within a single framework is appealing and potentially impactful.
3. The proposed approach is compatible with existing sparse SAE-based methods.

**Weaknesses:**

1. The rank requirement of the concept matrix implies that the number of concepts must exceed the number of intermediate activations. This condition may limit the practical scalability of the method in real-world scenarios.
2. The evaluation is conducted on a relatively small LLM (8B parameters), making it unclear whether the approach scales to larger foundation models commonly used in practice.
3. Experiments are limited to some specific LLM layer, and the rationale for this choice is not sufficiently discussed. It remains uncertain whether the findings generalize across different layers.

**Questions:**

1. Does the proposed framework introduce additional computational overhead compared with prior neuron-interpretation or behavior-control methods?
2. What are the key limitations of the proposed approach, particularly with respect to scalability and practical deployment?

---

> ### Author Response · Authors · 2025-11-23
> **Response to the reviewer rWXW (Part 1/2)**
>
> Thanks for appreciating our work and we would like to express our deepest gratitude to thank the reviewer for the constructive feedback!
>
>
> **Weakness 1: That the concept number exceeds activation's channel dimension might limit practical scalability.**
>
> (1) That the concept number exceeds activation's channel dimension is a subjectively existing mathematical constraint to guarantee a precise reconstruction, which applies to all interpretation frameworks interpreting neurons via decomposing it into a set of interpretable concepts.
>
>
> (2) Practically, users could leverage LLM to generate relevant concepts automatically or leverage trained SAE concepts to construct the concept matrix in our framework [line 258-259]. So users do not need to manually write a large number of concepts (but they can if they want).
>
> (3) The scalability is indeed a known problem of SAE frameworks because it is computationally expensive to learn a huge concept matrix. However, our unified framework also allows specifying concept matrix via the embeddings of conversations/sentences. This makes our framework rather scalable because it's quite straightforward to obtain e.g., 10000 concepts via conducting the feedforward process of a LLM for 10000 times and concatenate the embeddings of 10000 conversations/setences to obtain a concept matrix consisting of 10000 concepts.
>
>
>
>
> **Weakness 2: Whether the approach scales to larger models.**
>
> Our framework has a clear theoretical guarantee to achieve precise neuron/steering vector reconstruction, which is independent of the model size or architecture [line 359]. The empirically evaluated models up to the scale of 8B are commonly adopted model sizes in prior works for fair comparisons [ref1, ref2, ref3].
>
> [ref1] Robert Huben, et, al. Sparse autoencoders find highly interpretable features in language models. ICLR2023
>
> [ref2] Nina Rimsky, et, al. Steering llama 2 via contrastive activation addition. ACL2024.
>
> [ref3] Bruce W. Lee, et, al. Programming refusal with conditional activation steering. ICLR2025.
>
> Due to the limit of computation resources, we are unable to experiment with models larger than 8B. Compared to any empirical result in larger models, we believe a theoretical guarantee indicates a stronger evidence instead of a weaker evidence because empirical results can not offer any guarantee.
>
> **Weakness 3: Experiments conducted in some specific layers.**
>
> Our method is not limited by specific layers. As shown in [section 5.2, line 387], our experiments conducted for proving the same control effect as AS [ref1] in increasing the refusal rate of harmful queries apply our frameworks in all layers. So our findings generalize across layers thanks to the theoretical guarantee our framework offers. Other experiments conducted in specific layers follow the choice of unified approaches/relevant baselines for fair comparisons.
>
> [ref1] Bruce W. Lee, et, al. Programming refusal with conditional activation steering. ICLR2025.

---

> > ### Author Response · Authors · 2025-11-23
> > **Response to the reviewer rWXW (Part 2/2)**
> >
> > **Question 1: Does the framework introduce additional computational overhead compared with prior interpretation and control methods?**
> >
> > Compared to existing SAE based interpretation frameworks and steering vector based control approaches, the computational overhead is ignorable.
> >
> > **Compared to prior SAE interpretation frameworks:** (1) If users use sentences' embedding as concepts, the concept specification is rather direct by running the feedforward inference process c times (c is the number of concepts), while SAE requires significant computation resources for training a huge SAE network with lots of feedforword and backpropagation computations. (2) If users use SAE concepts to build the concept matrix, one needs to additionally calculate the $\mathbf{W}_{dec}$ via equation 9. The equation 9 consists of the matrix multiplication of $\mathbf{C}^T$ and $(\mathbf{C}\mathbf{C}^T)^{-1}$. Simple matrix multiplication is effective, the following table shows the computation time of calculating the $(\mathbf{C}\mathbf{C}^T)^{-1}$ using torch.inverse() library in randomly generated concept matrices corresponding to channel dimension $n=512,1024,2048,4096,8192$ to simulate different cases for different models with different channel numbers. The hardware is CPU: 10 vCPU Intel Xeon Processor (Skylake, IBRS).
> >
> > | channel number n| running time (seconds) |
> > |-------------|----------|
> > | 512| 0.01048|
> > |1024 |0.01234 |
> > |2048 | 0.05288 |
> > | 4096 | 0.30564 |
> > |8192 | 1.80868 |
> >
> > This computation only needs to be done once after the concept matrix is specified and there is no need to compute it again in inference time. So such a computational overhead in the seconds level is ignorable.
> >
> > **Compared to prior control approaches**: (1) For methods based on steering vector of the type 1, no additional computation is required because we can directly write the coefficients to reconstruct the steering vector as introduced in [line 313-317]. (2) For steering vector of the type 2, only a standard least squared optimization problem needs to be solved once [line 318-320]. The following table shows the computation time of executing this optimization using numpy.linalg.lstsq() under different channel dimension number and concept number using randomly generated concepts and steering vectors of corresponding shapes (n is the channel number and c is the cocnept number):
> >
> >
> > | | Computation time (seconds) | Steering vector reconstruction error |
> > |------|------|-------|
> > |n=512, c=1000| 0.16648 | $6.96\times 10^{-6}$|
> > |n=512, c=16384|0.307095| $1.15\times 10^{-5}$|
> > |n=1024, c=2000|0.04161|$4.64\times 10^{-6}$|
> > |n=1024, c=16384|0.618221|$1.36\times 10^{-5}$|
> > |n=4096,c=10000|1.5863|$1.93\times 10^{-5}$|
> > |n=8192,c=10000|3.1639|$2.67\times 10^{-5}$|
> > |n=8192,c=16384|5.1118|$3.45\times 10^{-5}$|
> >
> > Since the above reconstruction process only needs to be done once and the inference process can simply load the reconstructed steering vectors, the computational cost of this operation within seconds is very low.
> >
> >
> > **Question 2: Is there any key limitation regarding scalability and practical deployment?**
> >
> > Our framework is actually significantly more scalable than prior works in practical deployment: training a large scale SAE is very computationally expensive and it is hard to scale the SAE to super large scale while our framework allows a direct flexible concept specification without additional training. For example, it is computationally very expensive to train a SAE with 100,000 concepts in its concept matrix (e.g., requires long training time and large GPU memory in a large dataset), but it is rather efficient to simply feed 100,000 conversations to the model and obtain corresponding conversation representations in specified layers and concatenate them together to construct the concept matrix.
> >
> >
> > We hope the above responses could address the reviewer's concerns and sincerely hope the reviewer could consider raising the score. Thanks!

---

### Official Review · Reviewer_fPCv · 2025-11-01

**Soundness:** 3
**Presentation:** 3
**Contribution:** 3
**Rating:** 6
**Confidence:** 2

**Summary:**

This paper proposes a unified framework that bridges two previously separate research directions in large language models (LLMs): neuron interpretation and behavior control. Traditional neuron interpretation methods use sparse autoencoders (SAEs) to extract interpretable concepts but fail to effectively influence model behavior, while behavior control techniques steer model outputs through vector interventions without interpretability. The proposed framework connects these two perspectives, enabling both interpretable understanding and effective control of internal neuron representations. It allows flexible, user-friendly specification of semantic concepts, preserves model performance, and quantifies the contribution of each concept to the steering process. Overall, the work provides a principled foundation for designing interpretation methods that inherently incorporate controllability in LLMs.

**Strengths:**

The proposed framework appears novel, and its motivation is clearly articulated. The paper is generally well written, and Figure 2 provides a helpful visualization of the overall design, aiding readers' understanding of the main idea.

**Weaknesses:**

Some components of the proposed design could be explained in greater detail for clarity. Additionally, the paper lacks a discussion or comparison of computational complexity and runtime efficiency, which would help assess the practical feasibility of the approach.

**Questions:**

1. As mentioned in Appendix E, the concept matrix $C$ can be quite large (e.g., $16384 \times 512$). Computing its pseudo-inverse to obtain $W_{dec}$ (Eq. (9)) may be computationally expensive. Could this become a bottleneck in practice? If so, how to address or mitigate this issue?
2. In the lower half of Figure 2, the user-specified descriptions are passed through a language model to construct the concept matrix. Does this language model require specific training or fine-tuning, or can any pre-trained language model be used directly here?
3. Related to the above two questions, it would be helpful to provide more detail on how users specify the concept matrix $C$. The current description references prior works but does not sufficiently explain the step-by-step process of constructing $C$, which makes it difficult for readers unfamiliar with those references to follow.

---

> ### Author Response · Authors · 2025-11-23
> **Response to the reviewer fPCv**
>
> Thanks for appreciating our work and we would like to express our deepest gratitude to thank the reviewer for the constructive feedback!
>
> **Question 1 & Weakness 2: The concept matrix can be quite large. Could the computation cost of obtaining the pseudo inverse be a bottleneck?**
>
> The pseudo inverse computation consists of an effective matrix multiplication between $\mathbf{C}^T$ and $(\mathbf{CC}^T)^{-1}$ as well as the inverse of $\mathbf{CC}^T$.
> The computation cost of this inverse operation is very low as it does not depend on the concept number. In equation 9, the matrix that needs to be inversed is $\mathbf{C}\mathbf{C}^T$. Since $\mathbf{C}\in \mathbb{R}^{n\times c}$, where n is the channel dimension and c is the concept number,
> $\mathbf{C}\mathbf{C}^T \in \mathbb{R}^{n\times n}$. When $c=16384$ and $n=512$, $\mathbf{C}\mathbf{C}^T \in \mathbb{R}^{512\times 512}$, which is not a large matrix and its **size does not depend on the concept number $c$**. Note that the channel dimension $n$ is often not a large number. The time complexity of computing the inverse of a square matrix is $O(n^3)$ using typical approaches such as LU decomposition. The following table shows the average running time of 3 randomly generated concept matrix with the shape of several typical channel dimension number using torch.inverse() in our server with CPU: 10 vCPU Intel Xeon Processor (Skylake, IBRS).
>
> | channel number n| running time (seconds) |
> |-------------|----------|
> | 512| 0.01048|
> |1024 |0.01234 |
> |2048 | 0.05288 |
> | 4096 | 0.30564 |
> |8192 | 1.80868 |
>
> Note that such a computation only needs to be done once and the inference process can simply load the computed $\mathbf{W}_{dec}$. In the case of 16384 512-dimensional concepts, computing the inverse takes only around 0.01 second so it will not be the computation bottleneck in practice.
>
> **Question 2: Can any pre-trained language model be directly used to obtain concept matrix?**
>
> Yes. The language model does not require any specific training or fine-tuning because we are exactly trying to explain or control a model after it is trained (So are all SAE based interpretation frameworks and steering vector based behavior control approaches).
>
> **Question 3 & Weakness 1: More details on how users specify the concept matrix to facilitate readers.**
>
> (1) Users could specify the concept matrix via automatically discovered concepts such as from the SAE. Denote the concepts learned by SAE as $\mathbf{C}_{sae} \in \mathbb{R}^{n \times c}$,
> indicating the $c$ n-dimensional concepts [equation 2], simply setting the concept matrix in our framework to be the concept matrix discovered in SAE completes the specification process.
>
> (2) Users could also specify concept matrix via the embedding of conversations/sentences. The embeddings are obtained via feeding the specified conversations/sentences into the language model and take the last token or average token representations in the target layer as the embedding of that conversation/sentence. Whether choosing the last token or average token embedding depends on the choice of the control approach we want to unify [line 254-265] to guarantee the same control effect in the later control process. Why it makes sense to
> use the last token/average token representations to represent the sentence: this is an empirical choice due to the next token prediction training paradigm of LLM and the self-attention mechanism that captures the global information of all tokens. The fact that the existing control approaches adopt these representations to construct a steering vector do achieve effective behavior control empirically justifies the plausibility of this choice. After obtaining the n-dimensional embeddings of c such conversations/sentences, we simply concatenate them to construct a concept matrix $\mathbf{C}_{unified} \in \mathbb{R}^{n\times c}$.
>
>
> We hope the above responses could address the reviewer's concerns and sincerely hope the reviewer could consider raising the score. Thanks!

---

### Author Response · Authors · 2025-12-03
**Summary of the discussion period**

We provide a summary of our rebuttal here to facilitate the AC in the following reviewing process.
We have received **3 positive scores (reviewers fPCv, rWXW, ig9f) and 1 marginally negative score (dJSr)**.
Reviewers agree that our framework is novel (fPCv, ig9f), conceptually clean (dJSr), potentially impactful (rWXW) and holds significant academic merit (ig9f). The motivation is important and clearly articulated (fPCv, ig9f), the paper is well written (fPCv, rWXW) and the method exposes a more human facing control surface (dJSr).

The only reviewer that gives a marginally negative score (4, marginally below acceptance but would not mind if paper is accepted) have the following 3 concerns and we summarize our responses below.

***Concern 1 of reviewer dJSr: Concepts are raw activations of sentence and the sentences are likely not clean features.***\
(1) Sentence is actually a very natural way to express abstract concepts and what a "clean" feature means lacks rigorous definition in the literature.
Whether/how a concept can be further decomposed into subconcepts is very subjective. Our framework explicitly acknowledges the **subjective nature of the concept organization** in the framework design, which we believe is actually an important contribution.\
(2) Even if sentences are not "clean" features, a comparative use of sentences in our framework can also yield "clean" features (e.g., (A+B+C)-(A+B)=C).



***Concern 2 of reviewer dJSr: High reconstruction error in SAE baseline due to different design objective. We should not lean into it.***\
(1) From the aspect of interpretation, SAE is the **best possible baseline** we could compare with.\
(2) SAE intends to minimize the reconstruction error because this indicates how much information is not explained by the framework. We compare with SAE in the reconstruction error to justify our interpretability benefits.\
(3) Our additional design objectives such as user-friendly interpretation interface and control effect is not directly related to our lower reconstruction error.


***Concern 3 of reviewer dJSr: Evaluating more behaviors in AXBENCH.***
We address this concern via **evaluating 5 more behaviors in AXBENCH** to show our framework maintains the control effect as the original approach thanks to the theoretical guarantee our framework offers.


The rest 3 reviewers give us positive scores and we summarize the concerns addressed below.

**Major concerns of the Reviewer fPCv: Implementation details and analysis of the computational complexity and runtime.**
We offer the required details and analysis in the response and show the **computational cost is ignorable**.

**Major concerns of the Reviewer rWXW: Practical scalability, including concept number requirement, whether it scales to larger model, generalizes to different layers, as well as the computational overhead.**
We respond that users do not need to manually write all concepts, our framework is architecture/size/layer agnostic, refer to our existing experiments conducted in multiple layers and show the computational overhead is ignorable. So our framework is **practically rather scalable**.

**Major Concerns of the Reviewer ig9f: Why the L2 norm which offers many small values instead of sparse values yields interpretable solution.**
We point out that there is a **misunderstanding** because we are minimizing the difference between $\mathbf{C}^T$ and $\mathbf{W}_{dec}$ ,

instead of minimizing the weights of $\mathbf{W}_{dec}$. \
(We separate the above sentence into 2 lines since we observe an equation rendering problem in openreview preview window.)

We sincerely want to thank the reviewers, ACs and PCs for the efforts spent on our paper.
We believe our work delivers an important contribution to the community in bridging the gap between pure neuron interpretation and effective behavior control.
This is crucial to truly understand the model behavior via interpretable internal representations.

---

### Note · Authors · 2026-01-27

I have read and agree with the venue's withdrawal policy on behalf of myself and my co-authors.

---

### Meta-Review · Area_Chair_V7Fs · 2026-01-10

**Summary:**

Few concerns raised by the reviewers
* can we compare with SAEs given the objectives are different?
* the notion of concept is weakly grounded. the methods simply take raw activateion vectors instead of disentagled latent factors as concepts.
* evaluation is conducted on a relatively small LLM (8B paras), unclear if the approach scales to larger models
* The solution seems to be dependent on the user choice of concepts and specific W_dec, how do we know this is what the model is _actually_ doing?
* Why the objective should be selective? And why one solutions, out of infinitely many, chosen by the authors make sense?

**Reviewer Concerns:**

Examples of few main concerns that I believe weren't addressed convincingly are:
* can we compare with SAEs given the objectives are different?
* The solution seems to be dependent on the user choice of concepts and specific W_dec, how do we know this is what the model is _actually_ doing?
* Why the objective should be selective? And why one solutions, out of infinitely many, chosen by the authors make sense?

**Reviewer Scores:**

I don't think the scores would be have changed significantly to indicate a clear accept.

Since the paper is borderline, I read the paper myself and was not very aligned with a few arguments. I could see why reviewers weren't fully convinced and didn't suggest a clear acceptance.

---

### Decision · Program_Chairs · 2026-01-26

Reject